# MicroRNA-155-5p, Reduced by Curcumin-Re-Expressed Hypermethylated *BRCA1*, Is a Molecular Biomarker for Cancer Risk in *BRCA1*-methylation Carriers

**DOI:** 10.3390/ijms24109021

**Published:** 2023-05-19

**Authors:** Nisreen Al-Moghrabi, Maram Al-Showimi, Nujoud Al-Yousef, Lamya AlOtai

**Affiliations:** 1Cancer Epigenetics Section, Department of Molecular Oncology, King Faisal Specialist Hospital and Research Centre, Riyadh 11211, Saudi Arabia; malshowimi@kfshrc.edu.sa (M.A.-S.);; 2Department of Life Sciences, College of Science & General Studies, Al Faisal University, Riyadh 11533, Saudi Arabia

**Keywords:** miR-155, *BRCA1* methylation, curcumin, cancer-free carriers

## Abstract

Constitutional *BRCA1*-methylation is a cancer risk factor for breast (BC) and ovarian (OC) cancer. MiR-155, regulated by *BRCA1*, is a multifunctional microRNA that plays a crucial role in the immune system. The present study assessed the modulation of miR-155-5p expression in peripheral white blood cells (WBCs) of BC and OC patients and cancer-free (CF) *BRCA1*-methylation female carriers. Additionally, we investigated the potential of curcumin to suppress miR-155-5p in *BRCA1*-deficient breast cancer cell lines. MiR-155-5p expression was measured using a stem-loop RT-qPCR method. Gene expression levels were determined using qRT-PCR and immunoblotting. MiR-155-5p was more highly expressed in the *BRCA1*-hypermethylated HCC-38 and UACC-3199 BC cell lines than in the *BRCA1*-mutated (HCC-1937) and WT *BRCA1* (MDA-MB-321) cell lines. Curcumin suppressed miR-155-5p in the HCC-38 cells but not in the HCC-1937 cells via the re-expression of *BRCA1*. Elevated levels of miR-155-5p were detected in patients with non-aggressive and localized breast tumors and in patients with late-stage aggressive ovarian tumors, as well as in CF *BRCA1*-methylation carriers. Notably, IL2RG levels were reduced in the OC and CF groups but not in the BC group. Together, our findings suggest opposing effects of WBC miR-155-5p, according to the cell and cancer type. In addition, the results point to miR-155-5p as a candidate biomarker of cancer risk among CF-*BRCA1*-methylation carriers.

## 1. Introduction

MicroRNAs (miRNAs/miRs) are cancer-related molecules that regulate cellular mechanisms such as proliferation, differentiation, and oncogenesis [1]. These small molecules, 18 to 22 base pairs long, regulate the expression of targeted genes by binding to the 3-’untranslated region of mRNA. The host gene MIR155HG, which was originally identified as the B-cell Integration Cluster (BIC) gene, encodes the microRNA miR-155 [2]. It is processed from the 1500-bp noncoding primary miRNA-155 (pri-miR-155) transcript that spans exon 3 of the *BIC* gene. The pri-miR-155 is transcribed in the nucleus to a stem-loop precursor, pre-miR-155, which is further processed in the cytoplasm, generating the miR-155 duplex, miR-155-3p, and miR-155-5p [3,4]. MiR-155-5p is the more common and functionally dominant variant.

Females who carry germline *BRCA1* pathogenic mutations are at high risk of developing early onset aggressive BC and OC. *BRCA1* inactivation via DNA promoter methylation is an alternative mechanism involved in sporadic BC and OC carcinogenesis [5,6]. Both *BRCA1*-mutated and *BRCA1*-methylated neoplasms have similar pathological features [5,7]. Although mounting evidence suggests a link between constitutional *BRCA1* promoter methylation and the risk of BC and OC development [8,9,10,11,12,13,14,15,16,17], more research is needed to determine whether cancer-free *BRCA1*-methylation carriers are at an increased risk of cancer development in the same way that germline *BRCA1* mutation carriers are.

A critical difference between epigenetic and genetic alterations is the reversibility of the former process. Recently, we reported the demethylation and restoration of *BRCA1* gene expression in the highly *BRCA1*-hypermethylated HCC-38 and UACC-3199 cell lines via curcumin treatment. Treating the cells with curcumin, which is the active component of the herb Curcuma longa, reduced the DNA promoter methylation level and restored the expression of the BRCA1 protein through the upregulation of the demethylating enzyme TET1 [18].

The immune system plays a pivotal role in monitoring and preventing the development of cancer. A key player in the host’s immune response to cancer is T lymphocytes [19]. For a tumor to survive the immune response, it induces T cell apoptosis among T cells in peripheral blood lymphocytes as a mechanism of inhibiting anti-tumor activity [20]. The decreased expression of IL-2RG, a T-cell functional molecule essential for the proliferation and survival of T cells, is a trigger for T cell apoptosis [20]. Notably, data have shown that curcumin treatment prevents tumor-induced T cell apoptosis in tumor-bearing mice via the restoration of IL-2RG expression [21]. Furthermore, recent interesting data have suggested the intrinsic role of *BRCA1* in supporting the function of CD8^+^ T cells [22]. The abundance of circulating CD8^+^ T was significantly lower in *BRCA1*-mutation carriers than in controls.

MiR-155 is essential for T and B cell proliferation and maturation [23,24,25,26]. The increased production of T cell functional molecules in tumors with elevated miR-155 levels indicates an increased anti-tumor immune response [27], as the upregulation of miR-155 in breast tumors is associated with positive prognostic factors [27]. Individuals with early breast cancer had higher levels of miR-155 in their plasma and blood cells, indicating an immune response at the early stages of the disease [28]. Some studies, on the other hand, have found that higher miR-155 levels enhance the advancement of breast and other solid tumors, as it is implicated in cancer cell proliferation as well as the process of epithelial-mesenchymal transition (EMT) [29,30,31]. Additionally, higher miR-155 expression levels have been found in invasive and metastatic breast malignancies [29,32], with a link to triple-negative breast cancer [30,33]. Although the clinical importance of elevated miR-155 levels and tumor progression is debatable, increasing levels of serum miR-155 in breast cancer patients relative to healthy female controls could serve as a non-invasive diagnostic biomarker [34,35,36]. The *BRCA1* gene is a negative regulator of miR-155-5p. Mutant mouse embryonic stem cells harboring the R1699Q BRCA1 gene have been shown to have high amounts of miR-155-5p. Additionally, mutant *BRCA1* human cell lines express 50-times more miR-155-5p than wild-type *BRCA1* cell lines. The transient expression of wild-type *BRCA1* in mutant *BRCA1* HCC-1937 cells suppressed the expression of miR-155-5p [37]. Furthermore, results demonstrate a substantial link between lower levels of BRCA1 protein and the overexpression of miR-155-5p in *BRCA1* mutant human breast cancers [37]. However, the influence of *BRCA1* promoter methylation on miR-155-5p expression in *BRCA1*-methylated breast and ovarian malignancies has not been explored.

In the present study, we investigated the effect of *BRCA1* promoter methylation on the expression level of miR-155-5p in the highly *BRCA1*-methylated cell lines HCC-38 and UACC-3199. In addition, we assessed whether curcumin-upregulated BRCA1 can regulate the level of miR-155-5p in these cells. We also looked at the expression and clinical significance of miR-155-5p in WBCs from *BRCA1*-methylated breast and ovarian cancer patients, as well as cancer-free *BRCA1*-methylation female carriers.

## 2. Results

### 2.1. MicroRNA-155-5p Is Up-Regulated in BRCA1-Methylated Breast Cancer Cell Lines

To investigate the influence of *BRCA1* promoter methylation on miR-155-5p expression, we first aimed to assess the levels of miR-155-5p in *BRCA1*-methylated cell lines. To that end, we assessed miR-155-5p expression in the highly *BRCA1*-methylated breast cancer cells HCC-38 and UACC-3199. HCC-38 is a triple-negative breast cancer (TNBC) cell line, and UACC-3199 is an estrogen receptor-negative/progesterone receptor-negative (ER-/PR-) cell line; both cell lines show a basal-like gene expression pattern [38]. First, we measured the level of the primary miRNA-155 RNA (BIC mRNA) in the two cell lines together with the *BRCA1* mutant cell line HCC-1937, using the *BRCA1* wild-type cell line MDA-MB-231 as a control. The level of BIC mRNA was 2-fold higher in the mutant *BRCA1* cells compared to the wild-type *BRCA1* cells. However, BIC mRNA was significantly higher (30-fold) in the methylated *BRCA1* cell lines compared to the mutant and WT cell lines (Figure 1A). The mature version of miR-155 was then quantified in all four cell lines. Interestingly, miR-155-5p levels were 35-fold higher in the mutant cell line than in the wild type. Nevertheless, miR-155-5p levels were 400 and 600 times greater in HCC-38 and UACC-3199 cells, respectively, than in MDA-MB-231 cells (Figure 1B). These results indicate that miR-155-5p is significantly more abundant in methylated cells than in mutant cells, showing that the epigenetically altered *BRCA1* gene has a role in the upregulation of miR-155 in these cells.

### 2.2. miR-155-5p Correlates Negatively with the Endogenous Levels of BRCA1 Protein in the BRCA1-Methylated Cell Lines

Data have shown that the upregulation of miR-155-5p correlates with the loss of BRCA1 expression in *BRCA1* mutant cells [37]. To investigate whether the high level of miR-155-5p in the *BRCA1*-methylated cell lines correlated with reduced expression of BRCA1 in these cells, we measured the level of BRCA1 protein in the four cell lines (Figure 1C). The level of BRCA1 was 1.6-, 4.5-, and 13-fold lower in HCC-1937, HCC-38, and UACC-3199, respectively, as compared to MDA-MB-231 (Figure 1D). These results indicate that the high levels of miR-155-5p in the *BRCA1*-methylated cell lines correlate with the reduced endogenous levels of BRCA1 protein in these cells.

### 2.3. Curcumin Downregulates miR-155-5p in HCC-38 but Not in HCC-1937 via the Re-expression of BRCA1 Protein

It has been reported that transient expression of wild-type BRCA1 in HCC-1937 cells repressed miR-155-5p expression [37]. One of the differences between epigenetic and genetic alterations is the reversibility of the former process. In our previous study, we showed that mRNA and protein levels of BRCA1 were increased in the curcumin-treated HCC-38 cell line by reducing promoter methylation [18]. To assess whether the re-expression of BRCA1 via curcumin treatment can repress miR-155-5p, we treated the HCC-38 and HCC-1937 cell lines with curcumin for 6 days. BRCA1 was increased 1.6- and 2.2-fold at the mRNA and protein levels, respectively (Figure 2A,B), in the HCC-38 cell line. However, in HCC-1937, we found a 1.5-fold increase in the level of BRCA1 mRNA with no increase in the protein level (Figure 2C,D). Then, we measured the level of miR-155-5p in both cell lines. We found a 2.3-fold decrease in the level of miR-155-5p in the HCC-38 cells at the 7.5 µM dose, but no decrease was observed at either 7.5 or 10 µM in the HCC-1937 cells, compared to the control (Figure 2E,F, respectively). These results indicate that curcumin downregulated miR-155-5p in the *BRCA1*-methylated cells but not in the mutant cells via the re-expression of the BRCA1 protein.

### 2.4. High miR-155-5p Expression Level in WBCs Is Associated with Favorable Prognostic Factors in BC Patients

Several studies have determined the clinical relevance of plasma miR-155 in BC patients [27,28]. We sought to evaluate the expression levels and the clinical significance of miR-155-5p in peripheral WBCs in patients with BC, including *BRCA1*-methylation-positive patients. To this end, we measured miR-155-5p in WBCs RNA from 46 randomly selected BC patients (median age 47), of whom 29 were positive for methylated *BRCA1* (Table 1); these data were obtained from the department of oncology in King Faisal Specialist Hospital and Research Centre (Riyadh, Saudi Arabia). The results revealed an overall highly significant increase in the expression levels of miR-155-5p in patients compared to controls (*p* = 0.0003) (Figure 3A). The highest level was observed in the subtype Luminal A (*p* ≤ 0.0001), which is a low-risk group, followed by Luminal B (*p* = 0.0028), which is of a higher grade and has a higher proliferative rate, followed by TNGBC (*p* = 0.0668), which is an aggressive form of BC (Figure 3B). Comparison among the subtype groups evidenced a statistically significant difference only in the Luminal A subtype compared with TNGBC (*p* = 0.0378) (Figure 3B). Furthermore, significantly higher levels of miR-155-5p were found in patients with localized tumors (*p* = 0.00021) and in patients with lymph node involvement (*p* = 0.00021) than in those with distant metastasis (*p* = 0.2783), compared to controls (Figure 3C). Overall, these findings suggest that high peripheral WBC miR-155-5p levels may act as a tumor suppressor in BC.

### 2.5. High miR-155-5p Expression Level in WBCs Is Associated with Favorable Prognostic Factors in BRCA1-Methylation-Positive BC Patients

Analyzing the 29 *BRCA1*-methylation-positive breast cancer patients separately revealed similar results to those of the overall group of BC patients, where the expression of miR-155-5p was significantly higher in patients compared to controls (*p* = 0.004) (Figure 3D). Furthermore, high levels of miR-155-5p were found in the non-TNGBC subtype (*p* = 0.0079), which was not the case for TNGBC (*p* = 0.143), with no difference between the two groups (Figure 3E). Again, higher levels of miR-155-5p were observed in patients with localized BC tumors (*p* = 0.0170) and in patients with lymph node involvement (*p* = 0.0509) than in patients with distant metastases (*p* = 0.1663), as compared to controls (Figure 3F).

### 2.6. High miR-155-5p Expression Level in WBCs Is Associated with Unfavorable Prognostic Factors in OC Patients

Little information is known about the clinical relevance of miR-155-5p in patients with OC. Thus, we sought to evaluate the expression levels and clinical significance of miR-155-5p in peripheral WBCs in patients with ovarian cancer, including *BRCA1*-methylation-positive patients. To this end, we measured miR-155-5p in WBC RNA from 44 randomly selected OC patients, of whom 20 were positive for methylated *BRCA1* (Table 2). Similar to BC patients, the results revealed an overall significant increase in the expression levels of miR-155-5p compared to controls (*p* = 0.003) (Figure 4A). However, higher levels were found in stages III–IV (*p* = 0.004) than in stages I-II (*p* = 0.156) (Figure 4B). Consequently, miR-155-5p was higher in patients with disease progression (*p* = 0.0002) than in patients with a stable disease (*p* = 0.052) compared to controls. A comparison between the two patient groups revealed a statistically significant difference (*p* = 0.048) (Figure 4C). Overall, these results indicate that, unlike breast cancer, high peripheral WBC miR-155-5p may be oncogenic in OC.

### 2.7. High miR-155-5p Expression Level in WBCs Is Associated with Unfavorable Prognostic Factors in BRCA1-Methylation-Positive OC Patients

Analyzing the 20 *BRCA1*-methylation-positive ovarian cancer patients separately revealed similar results to the whole group of OC patients, where the expression of miR-155-5p was significantly higher in patients as compared to controls (*p* = 0.043) (Figure 4D). Again, patients with disease progression had a higher level of miR-155-5p (*p* = 0.032) than those with stable disease (*p* = 0.420) (Figure 4E). Altogether, these results reveal that high peripheral WBC miR-155-5p acts as a tumor suppressor in BC and as an oncogene in OC, regardless of the methylation status of *BRCA1*.

### 2.8. IL2RG Level Is Reduced in Ovarian Cancer Patients with high WBC miR-155-5p

Data have shown that the levels of T cell functional molecules were significantly elevated in high miR-155 tumors, revealing enhanced anti-tumor immunity within these tumors [27]. To investigate the effect of high WBC miR-155-5p levels on T cell functional molecules, we assessed the expression level of IL2RG in patients’ WBCs compared to controls. The results revealed an insignificant increase in the IL2RG level in patients with breast cancer compared to controls (*p* = 0.657) (Figure 5A). However, there was a significant decrease in the level of IL2RG in patients with ovarian cancer (*p* = 0.020) (Figure 5B). Interestingly, comparison between the two cancer types revealed a highly significant reduction in the level of IL2RG in patients with ovarian cancer compared to those with breast cancer (*p* = 0.0027) (Figure 5C). Our findings may indicate that, in patients with OC, high WBC miR-155-5p may not be associated with enhanced antitumor immunity.

### 2.9. miR-155-5p Is Elevated in WBCs of BRCA1-Methylation Female Carriers

To evaluate the use of WBC miR-155-5p as a molecular biomarker for the assessment of cancer risk in CF-*BRCA1* methylation-carrying females, we measured the level of WBC miR-155-5p in ten *BRCA1*-methylation carriers, as used in our previous study [39]. Notably, similar to the *BRCA1*-methylation-positive patients with breast and ovarian cancers, miR-155-5p was significantly higher in the *BRCA1*-methylated carriers compared to age-matched controls (*p* = 0.0418) (Figure 6A). Furthermore, as *BRCA1* promoter methylation is evident in carriers from early on in life [17], we sought to assess the expression of miR-155-5p in newborn female carriers. To this end, we measured miR-155-5p in the WBCs of 13 *BRCA1*-methylation newborn female carriers used in our previous study [39]. Notably, miR-155-5p was significantly higher in the newborn carriers compared to newborn controls (*p* = 0.026), with two samples showing huge increases, 30- to 56-fold higher (Figure 6B). These results suggest that CF-*BRCA1* methylation carriers have altered WBC miR-155-5p changes similar to those seen in patients with breast and ovarian cancer early in life.

### 2.10. IL2RG Level Is Decreased in WBCs of CF-BRCA1-Methylation Carriers

To investigate the effect of the high WBC miR-155-5p levels on the T cell functional molecules in CF-*BRCA1*-methylated carriers, we measured the expression level of WBC IL2RG in the adult carriers. The results revealed a statistically significant reduction in IL2RG in the carriers as compared to controls (*p* = 0.0012) (Figure 6C). This result may imply a reduced antitumor immunity in those individuals, indicating an elevated susceptibility to cancer risk.

### 2.11. High miR-155-5p in BRCA1-Methylated WBCs Is not Associated with Reduced Endogenous BRCA1

We have shown above that high miR-155-5p levels correlate negatively with the endogenous levels of BRCA1 protein in the *BRCA1*-methylated cell lines. To investigate whether high WBC miR-155-5p is associated with a reduced level of BRCA1 in *BRCA1*-methylated WBCs, we assessed the level of BRCA1 mRNA in WBCs from breast and ovarian *BRCA1*-methylation-positive patients as well as CF-*BRCA1*-methylated carriers. A comparison of the pairwise expression levels between miR-155-5p and mRNA BRCA1 within WBCs for each patient and carrier revealed a significant negative correlation (*p* = 0.0001 and *p* = 0.034) (Figure 7A,B) for patients with breast and ovarian cancer, respectively, with marginal significance (*p* = 0.055) for the carriers (Figure 7C), which was not the case for controls. On the other hand, a comparison of the level of BRCA1 mRNA between patients and controls revealed no difference between the two groups. However, a significant increase in the level of the BRCA1 mRNA was observed in the carriers (*p* = 0.015) (Figure 7C) compared to age-matched controls. These results indicate that, unlike cancer cell lines, a high WBC miR-155-5p is not associated with reduced endogenous levels of BRCA1.

## 3. Discussion

MiR-155 is a key player in the proliferation and maturation of immune cells and antitumor immunity [23,24,25,26,40]. Additionally, high serum miR-155-5p could serve as a biomarker in the diagnosis and prognosis of BC [34,35,36]. Due to its versatile functions, it is uncertain whether miR-155-5p is oncogenic or tumor suppressive [29,30,31]. As *BRCA1* is a negative regulator of this puzzling microRNA [37], it is appropriate to test the clinical significance of miR-155 in *BRCA1*-methylated human cells.

In the present study, our analysis detected significantly augmented endogenous levels of miR-155-5p in HCC-38 (Claudin-low) and UACC-3199 (basil-like), which are aggressive TNBC cell lines with a highly hypermethylated *BRCA1* promoter. These results are comparable to those in the TNBC *BRCA1* pathogenic variant cell line MDA-MB-436 (basal-like). These findings suggest a role for the epigenetically altered *BRCA1* in the negative regulation of miR-155-5p similar to the pathogenic *BRCA1* mutation [31]. Reported data have shown that the loss of wild-type *BRCA1* or the loss of its function correlates with the upregulation of miR-155-5p [37]. In the present study, we found a strong negative correlation between the level of miR-155-5p and the endogenous level of *BRCA1* in the four tested cell lines. Notably, the re-expression of BRCA1 protein in the HCC-38 cells via curcumin treatment, in agreement with our previous study [18], reduced the level of miR-155-5p. Curcumin had no influence on the level of miR-155-5p in the HCC-1937 cell line, despite a slight increase in *BRCA1* mRNA but not BRCA1 protein. Reported data have shown that the expression of miR-155-5p in the mutant *BRCA1* HCC-1937 cells was repressed by the transient expression of wild-type *BRCA1* in these cells [37]. Our data suggest that the reduced miR-155-5p level in the curcumin-treated HCC-38 is due to the re-expression of functional *BRCA1* and not to curcumin per se. The fact that curcumin-treated HCC-38 cells have lower proliferation rates than untreated cells [18] demonstrates the oncogenic effect of miR-155-5p in these cells.

On the other hand, our present findings reveal that high WBC miR-155-5p is associated with favorable prognostic factors in patients with BC, including *BRCA1*-methylation-positive patients. These findings are in line with previously reported data revealing higher levels of miR-155 in plasma and peripheral WBCs in patients with localized tumors than in those with distant metastases, compared to healthy female controls [28]. These results may indicate that the function of miR-55-5p in tumor cells differs from that in peripheral WBCs.

In line with this notion, increased miR-155-5p in ovarian cancer cell lines has been reported to prevent ovarian cancer tumorigenesis, inhibit cell proliferation, and enhance apoptosis [4,41,42], suggesting a tumor-suppressive effect of miR-155-5p in these cells. Based on our findings, we reveal that high WBC miR-155-5p is associated with unfavorable prognostic factors in patients with OC, including *BRCA1*-methylation-positive patients, as WBC miR-155-5p was higher in patients with disease progression compared to controls. These results reveal that WBC miR-155-5p has dual functions, being tumor suppressive in BC but oncogenic in OC. These data are in line with the results obtained for miR-126 in our previous study, where we showed that high WBC miR-126 was associated with a lower risk of distant metastasis in BC but a higher risk of disease progression and death in OC [39]. The dual functions of these miRs may indicate cancer-type-dependent mechanisms of action. However, further studies are needed to explore the dual functions of these miRs.

Considering the immune regulatory role of miR-155, it is implied that higher miR-155 levels can indicate efficient antitumor immunity. The increased expression of T cell functional molecules in miR-155-high tumors is an indication of enhanced anti-tumor immunity [27]. Our findings show that, although there was an insignificant increase in IL2RG in patients with breast cancer compared to controls, the level of this gene was significantly reduced in patients with ovarian cancer compared to controls and patients with breast cancer. The decreased expression of IL2RG induces T cell apoptosis among T cells in the peripheral blood, resulting in lowered anti-tumor immunity. This phenomenon may explain the oncogenic effect of miR-155-5p in patients with ovarian cancer. However, further studies are needed to explore the link between WBC miR-155-5p and anti-tumor immunity in ovarian cancer.

ILR2G was also considerably reduced in the CF-*BRCA1* methylation carriers compared to age-matched controls, despite the high level of WBC miR-155-5p. This result suggests lower anti-tumor immunity in those carriers. A recent report has shown that altered *BRCA1* expression in peripheral T cells could result in aberrant transcription related to antitumor immunity, which may contribute to the elevated breast cancer risk among *BRCA1*-mutation-carrying women [22]. Based on these data, it is plausible to argue that, similar to germline *BRCA1* mutations, constitutional *BRCA1*-methylation may likewise increase the risk of cancer formation in *BRCA1*-carrying women. This notion is supported even further by our recent findings, where we have shown that WBC miR-126 levels were considerably higher in CF *BRCA1*-methylation carriers than in age-matched controls, which was consistent with the findings in patients with BC and OC [39]. The upregulation of WBC miR-126 is deemed to be a sign of constitutional *BRCA1* promoter methylation-related-malignancies.

Our results revealed a significant negative correlation between the endogenous levels of miR-155-5p and mRNA *BRCA1* within WBC for patients with breast and ovarian cancer and *BRCA1*-methylation carriers, which is not the case in controls. However, the increase in WBC miR-155-5p in those individuals was not associated with reduced *BRCA1* expression compared to controls. Evidence suggests that the increased plasma and blood cells miR-155-5p is an immunologic response during the early stages of breast cancer [27], which might imply a *BRCA1*-independent response.

In conclusion, the current study revealed a significant increase in WBC miR-155-5p in patients with breast and ovarian cancer, which has opposing pathogenic effects according to cell type and cancer type. CF-*BRCA1*-methylation-carrying women appear to have lower antitumor immunity, indicating an elevated cancer risk among those individuals. The fact that curcumin is a suppressor of miR-155 in tumor cells and a restorer of IL2RG in blood cells [21] holds the potential to be an effective preventive therapy against cancer formation in *BRCA1*-metylation carriers.

## 4. Materials and Methods

### 4.1. WBC DNA and RNA

In this work, we used leftover samples for WBC DNA and RNA from our previous study [39]. In the case of BC, 46 samples were used (ages 31 to 82; median age 47 years), 29 of which were positive for methylated *BRCA1.* Furthermore, 44 OC samples were used (age range, 19–88 years; median age, 52 years), 20 of which were positive for methylated *BRCA1*. In addition, 30 female cancer-free samples (age range 17–53 years), 10 of which were positive for methylated *BRCA1*, were used and 18 newborn female WBC RNA samples from our previous work were also utilized [17], 10 of which were positive for *BRCA1* methylation.

### 4.2. Cell Culture and Treatment

The HCC-38, UACC-3199, HCC-1937, and MDA-MB-231 breast cancer cell lines were obtained from the American Type Culture Collection (ATCC, Manassas, VA, USA). The cells were cultured in RPMI-1640 medium supplemented with 10% FBS, 100 U/mL penicillin, and 100 g/mL streptomycin (Gibco/Life Technologies, Thermo Fisher Scientific, Inc, Waltham, MA, USA). The cells were treated with 7.5 and 10 µM curcumin (Sigma-Aldrich; Merck KGaA, Burlington, MA, USA) at 40–60% confluence and incubated for 6 days in a humidified atmosphere at 37 °C and 5% CO_2_. Then, the cells were collected for RNA and protein extraction.

### 4.3. RT-qPCR

cDNA was generated from RNA using Superscript III, reverse transcriptase, and random hexamers (Applied Biosystems; Thermo Fisher Scientific, Inc., Waltham, MA, USA; cat. no. 4368814). qPCR was performed using specific primers for the BRCA1, BIC, and IL2RG transcripts (Table 3), using actin as a housekeeping gene. All the primers and the qPCR conditions are stated in Table 3. PCR was performed using the CFX96 Real-Time System (Bio-Rad Laboratories, Inc., Hercules, CA, USA, USA) with SYBR Green (RT^2^ SYBR Green Fluor qPCR Mastermix; cat. no. 330513; Qiagen GmbH, Hilden, Germany). The 2−^ΔΔCq^ method was used to calculate the relative expression of the measured mRNAs. For the cells, the fold change of BRCA1 expression in curcumin-treated cells was measured relative to DMSO-treated cells. For WBC, the fold change of BRCA1 mRNA expression in patients and *BRCA1*-methylation carriers was performed relative to age-matched controls.

### 4.4. Stem-Loop PCR Assay

RT-qPCR for miR-155-5p was performed using a stem-loop RT primer and TaqMan miRNA RT kit (miRBase ID: hsa-miR-155-5p; catalog no. 4440886; Applied Biosystems; Thermo Fisher Scientific, Inc.) following the manufacturer’s protocols. The primer and the thermo cycling condition are given in Table 3. The small nuclear RNA U6 (U6; assay ID: 001973) was used for normalization (Table 3). The expression level was calculated based on the threshold cycle value using the 2−^ΔΔCq^ method [43]. For the cells, the fold change of miR-155-5p expression was performed relative to DMSO-treated cells. For WBCs, the fold change was measured relative to age-matched controls.

### 4.5. Western Blot Analysis

Protein extraction from the cells was carried out using a RIPA lysis buffer (R0278; Sigma-Aldrich; Merck KGaA). Protein quantification was carried out using the Bradford method. Protein (50 µg) was subjected to 10 and 12% SDS-PAGE, which was then transferred to PVDF membranes. After blocking at room temperature with 5% non-fat milk for 1 h, the membranes were incubated at 4 °C overnight with primary antibodies (dilution 1:1000) BRCA1 (ab9141) (purchased from Abcam, Cambridge, UK) and GAPDH (purchased from Cell Signaling Technology, Inc., Danvers, MA, USA). The membranes were visualized using ECL Detection reagents (Pierce; Thermo Fisher Scientific, Inc., USA). Images were visualized using a LAS-4000 Imager (Fujifilm, Tokyo, Japan). Band quantification was performed using the GelQuant.NET program (version 1.8.2; BiochemLab, San Francisco, CA, USA).

### 4.6. Statistical Analysis

An unpaired *t*-test was used to determine the statistical significance between two groups, curcumin-treated and untreated cells, between adult CF carriers vs. controls and between newborn carriers vs. newborn controls. A one-way ANOVA with Dunnett’s multiple comparison tests was performed for comparing multiple groups. GraphPad version 9.1.0 (GraphPad Software, Inc., La Jolla, CA, USA) was used for all analyses. *p* < 0.05 was considered as indicating a statistically significant difference.

## Figures and Tables

**Figure 1 ijms-24-09021-f001:**
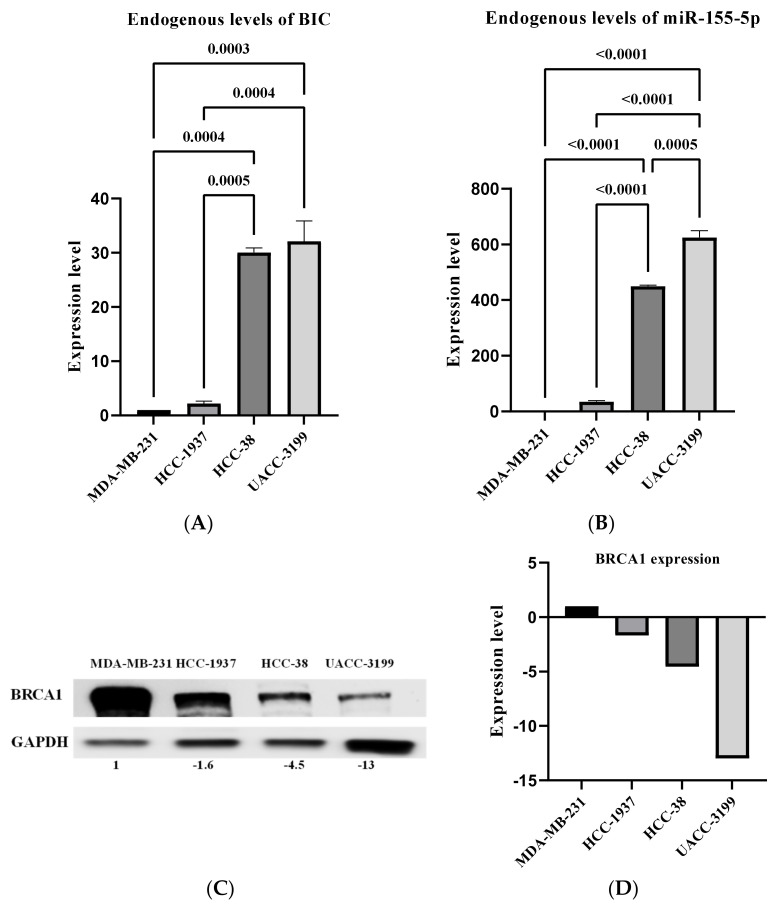
The expression of miR-155 and BRCA1 protein in breast cancer cell lines. The expression of primary miRNA-155 RNA (BIC) was measured using RT-qPCR. The expression of miR-155-5p was measured using stem-loop RT-qPCR. The expression of BRCA1 protein was measured using an immunoblotting assay. (**A**) The analysis of BIC in breast cancer cell lines. (**B**) The analysis of miR-155-5p in breast cancer cell lines. (**C**) The expression of the BRCA1 protein in breast cancer cell lines using GAPDH as an internal control. (**D**) The quantitative analysis of the expression levels of BRCA1 protein in the four cell lines.

**Figure 2 ijms-24-09021-f002:**
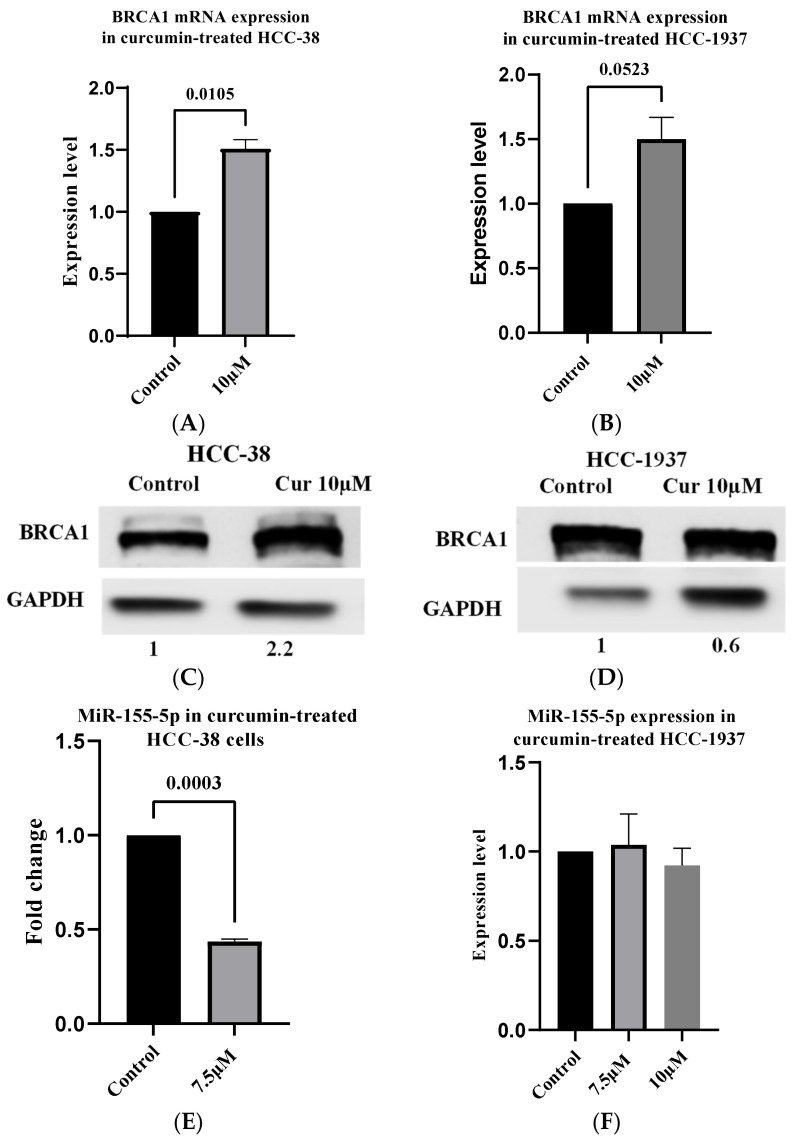
Curcumin induces the expression of BRCA1 and reduces miR-155-5p in HCC-38 but not HCC-1937. The expression of BRCA1 mRNA was measured using RT-qPCR. The expression of miR-155-5p was measured using stem-loop RT-qPCR. The expression of BRCA1 protein was measured using an immunoblotting assay. The cells were treated with 7.5 and 10 µM curcumin for 6 days, and the effects of curcumin on BRCA1 mRNA and protein expression in (**A**,**C**) HCC-38 and (**B**,**D**) HCC-1937 cells, respectively, are shown. (**E**) The analysis of miR-155-5p in HCC-38 cells. (**F**) The analysis of miR-155-5p in HCC-1937 cells. Control cells were treated with dimethyl sulfoxide. Error bars represent the mean ± SD. Cur—curcumin.

**Figure 3 ijms-24-09021-f003:**
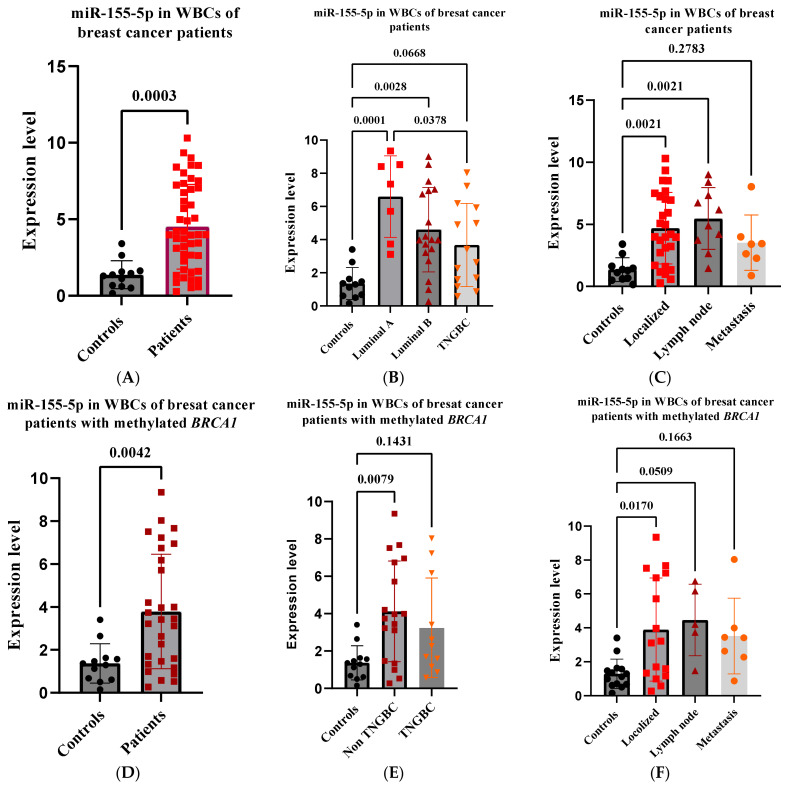
Expression of miR-155-5p in peripheral WBCs in patients with breast cancer. The expression of miR-155-5p was measured using stem-loop RT-qPCR. (**A**–**C**) The analysis of miR-155-5p expression in WBCs of patients with breast cancer. (**D**–**F**) The analysis of miR-155-5p expression in WBCs of patients with *BRCA1*-methylated breast cancer. Error bars represent the mean ± SD. RT-qPCR—reverse transcription-quantitative polymerase chain reaction; WBC—white blood cells.

**Figure 4 ijms-24-09021-f004:**
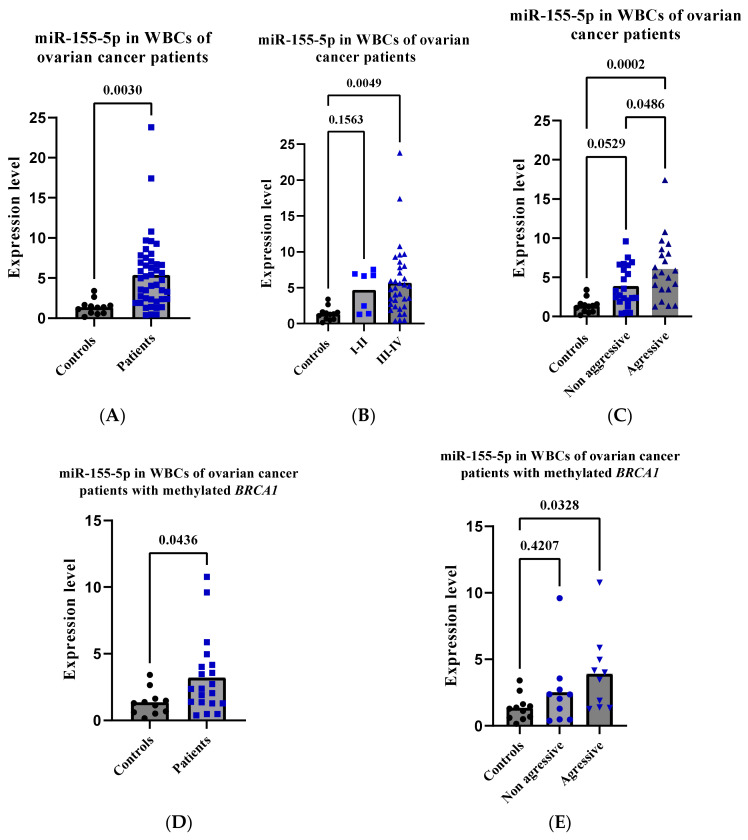
Expression of miR-155-5p in peripheral WBCs in patients with ovarian cancer. The expression of miR-155-5p was measured using stem-loop RT-qPCR. (**A**–**C**) The analysis of miR-155-5p expression in WBCs of patients with ovarian cancer. (**D**,**E**) The analysis of miR-155-5p expression in WBCs in patients with *BRCA1*-methylated ovarian cancer. RT-qPCR—reverse transcription-quantitative polymerase chain reaction; WBC—white blood cells; I–II—low stages; III–IV—high stages.

**Figure 5 ijms-24-09021-f005:**
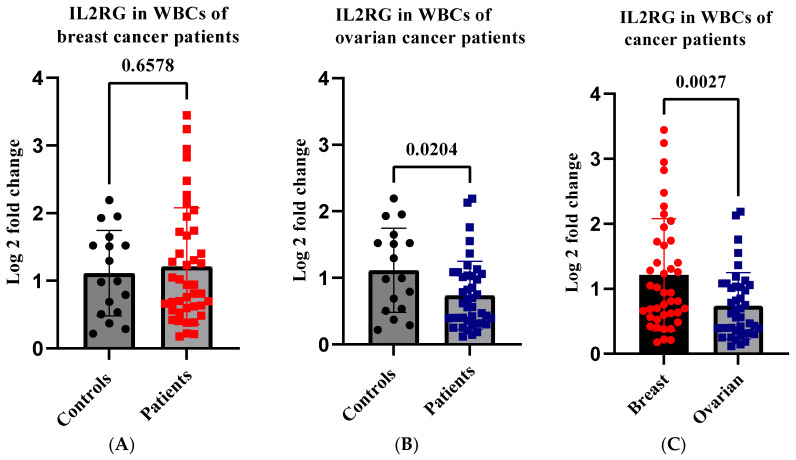
Expression of IL2RG in peripheral WBCs in patients with breast and ovarian cancer. IL2RG mRNA was measured using RT-qPCR. (**A**) The analysis of IL2RG mRNA in the WBCs of patients with breast cancer. (**B**) The analysis of IL2RG mRNA in the WBCs of patients with ovarian cancer. (**C**) The comparison of IL2RG mRNA levels between breast and ovarian cancer. Error bars represent the mean ± SD. RT-qPCR—reverse transcription-quantitative polymerase chain reaction; WBC—white blood cells.

**Figure 6 ijms-24-09021-f006:**
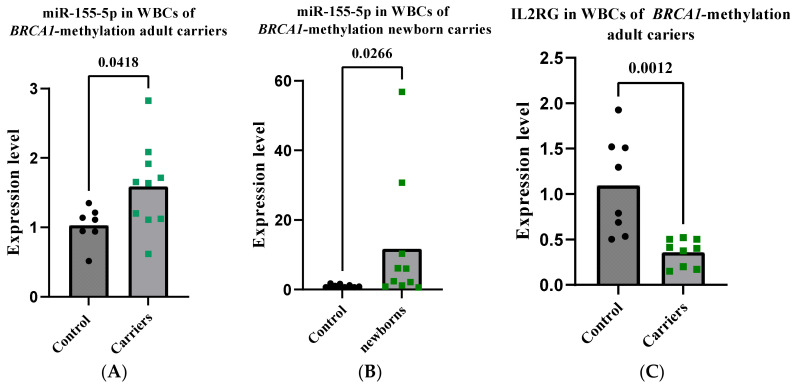
Expression of miR-155-5p and IL2RG in peripheral WBCs in *BRCA1*-methylation-female carriers. The expression of miR-155-5p was measured using stem-loop RT-qPCR. IL2RG mRNA was measured using RT-qPCR. (**A**) The analysis of miR-155-5p expression in WBCs of cancer-free *BRCA1*-methylation adult carriers. (**B**) The analysis of miR-155-5p expression in WBCs of *BRCA1*-methylation newborn carriers. (**C**) The analysis of IL2RG mRNA in the WBCs of cancer-free *BRCA1*-methylation adult carriers. Error bars represent the mean ± SD. RT-qPCR—reverse transcription-quantitative polymerase chain reaction; WBC—white blood cells.

**Figure 7 ijms-24-09021-f007:**
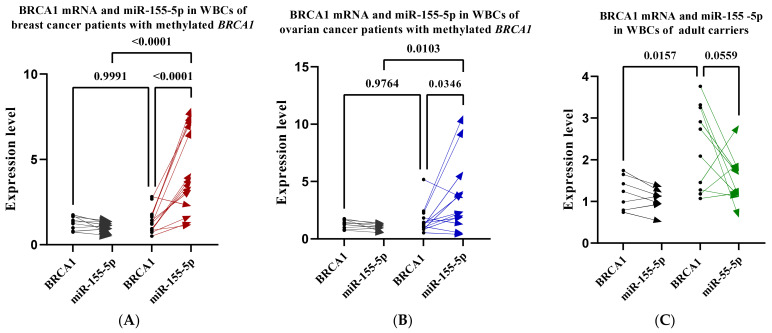
The expression of miR-155-5p was measured using stem-loop RT-qPCR. BRCA1 mRNA was measured using RT-qPCR. (**A**) Pairwise expression levels between miR-155-5p and mRNA BRCA1 within WBCs for controls (black arrows) and patients with *BRCA1*-methylated breast cancer (red arrows). (**B**) Pairwise expression levels between miR-155-5p and mRNA BRCA1 within WBCs for controls (black arrows) and patients with *BRCA1*-methylated ovarian cancer (blue arrows). (**C**) Pairwise expression levels between miR-155-5p and mRNA BRCA1 within WBCs for controls (black arrows) and cancer-free BRCA1-methylation adult carriers (green arrows).

**Table 1 ijms-24-09021-t001:** Clinicopathological features of breast cancer patients.

Features	*n* = 47
*BRCA1* Meth	29
IDC	27
TNG	15
DCIS	3
Others	2

Meth—methylated; IDC—invasive ductal carcinoma; TNG—triple negative; DCIS—ductal carcinoma in situ.

**Table 2 ijms-24-09021-t002:** Clinicopathological features of ovarian cancer patients.

Features	*n* = 44
*BRCA1* Meth	20
HGSOC	32
Others	12

Meth—methylated; HGSOC—high grade serious ovarian cancer.

**Table 3 ijms-24-09021-t003:** RT-quantitative PCR primers.

Primer Name	Primer Sequence	Annealing Temp
*RT BRCA1*	F5′-TGTAGGCTCCTTTTGGTTATATCATTC–3′ R5′-CATGCTGAAACTTCTCAACCAGAA–3′	59 °C
IL2RG	F5′-GTGCTCAGCATTGGAGTGAA R5′-CCCGTGGTATTCAGTAACAAGA	59 °C
β-Actin	F5′-TCC CTG GAG AAG AGC TAC GA–3′ R5′-TGA AGG TAG TTT CGT GGA TGC–3′	59 °C
miR-155-5p Stem-loop	CUGUUAAUGCUAAUCGUGAUAGGGGUUUUUG CCUCCAACUGACUCCUACAUAUUAGCAUUAACAG	60 °C
U6	GTGCTCGCTTCGGCAGCACATATACTAAAATTGGAA CGATACAGAGAAGATTAGCATGGCCCCTGCGCAAG GATGACACGCAAATTCGTGAAGCGTTCCATATTTT	60 °C

## Data Availability

All data generated or analyzed during this study are available from the corresponding author on reasonable request.

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
