# Peer review of "MicroRNA-155-5p, Reduced by Curcumin-Re-Expressed Hypermethylated BRCA1, Is a Molecular Biomarker for Cancer Risk in BRCA1-methylation Carriers"

_ijms, 2023, doi:10.3390/ijms24109021_

Round 1
Reviewer 1 Report
Review

Author Response
Response to Reviewer 1 Comments Point 1: What is the difference between the BRCA1-hypermethylated HCC-38 and UACC3199 BC cell lines? Response 1: HCC-38 is a triple-negative breast cancer (TNBC) cell line, and UACC-3199 is an estrogen receptor-negative/progesterone receptor-negative (ER-/PR-) cell line; both cell lines show a basal-like gene expression pattern (A refrence is added to the result section (38) Point 2: 2.1. microRNA-155-5p is up-regulated in BRCA1-methylated breast cancer cell lines Could you add a bit more information about the BIC mRNA, like primary-miRNA-155 RNA. You should mention this form of the name in the text part as well.. Response 2: Information about BIC mRNA is already mentioned in the introduction. The name primary-miRNA-155 RNA is added to the result (highlited in yelow). Point 3: There are two Figure 1. Response 3: The figure number is corrected. Point 4. 2.1.4. High miR-155-5p expression level in WBCs is associated with favorable prognostic factors in 212 BC patients You should give in this part the median age of the patients and the name of the hospital from which the samples originate. Could you include short information about the Luminal A, Luminal B and TNGBC subtype Response 4. Median age and the name of the hospital from which the samples originate are added. Information about the Luminal A, Luminal B and TNGBC subtype is added. Point 5. The Table 2. should be the Table 1 Response 5. All table numbers are changed Point 6. 2.1.8. IL2RG level is reduced in ovarian cancer patients with high WBCs miR-155-5p. In this part you mention, that there is no differences in the IL2RG level between patients with breast cancer and controls. In the patients the expression level of IL2RG should not be higher than in the control group? Response 6. In fact, the results show a slight but insignificant increase in IL2RG in breast cancer patients compared to controls. The statement has been changed to “The results revealed an insignificant increase in the IL2RG level in patients with breast cancer compared to controls” in the result and discussion sections. Point 7. 2.1.11. High miR-155-5p in BRCA1-methylated WBCs is not associated with reduced endogenous 452 BRCA1. In this part you give the information that unlike cancer cell lines, high WBCs miR155-5p are not associated with reduced endogenous levels of BRCA1. What could be the reason of the difference? Response 7. In the cell lines, there is a significant decrease in the level of BRCA1 protein as the two alleles are affected due to a douple hit, which is not the case in the WBCs. Also, as already stated in the discussion section, "Evidence suggests that the increased plasma and blood cells miR-155-5p are an immunologic response during the early stages of breast cancer [27], which might imply a BRCA1-independent response."”. Point 8. In the discussion part could you give the name some target genes of miR-155-5p to prove that it could function as tumor supressor or oncomiR as well. Response 8. In WBCs, the function of miR-155-5p as a tumor suppressor or oncomiR is through its function as a regulator of the immune system and not as a direct action on specific genes. Point 9. Please give more detaled information in the part of statistical analysis Response 9. What more details should I include? English editing is done
Reviewer 2 Report
The manuscript entitled “MicroRNA-155-5p, reduced by curcumin-re-expressed hyper-2 methylated BRCA1, is a molecular biomarker for cancer risk in 3 BRCA1-methylation carriers” by Moghrabi. et al., have looked at the expression and clinical significance of miR-155-5p in breast and ovarian cancer patients as well as BRCA1-methylated carriers. The reviewer thinks this is an important contribution to the field; however, there are several questions that needs to be addressed. The major concern is the lack of functional experiments.
Major Points
1. There are several studies on the involvement of miR-155-5p in breast cancer and other tumors, the authors should emphasize the gap in the field and how this research is an advancement over the existing literature.
2. The statement “Overall, these results indicate that high peripheral WBCs miR-155-5p may be tumor suppressive in BC” is not convincing to the reviewer. No functional experiments have been performed. The oncogenic and tumor suppressor role of miR-155-5p in breast cancer have been a topic of debate. Additionally, the number of breast cancer samples used in the study is not large enough for a conclusive statement. The reviewer would advise the authors to use TCGA or other publicly available datasets, segregate patients by BRCA1 status (WT, mutated and methylated) and repeat the analysis. Cox regression and survival analysis should reveal the importance of miR-155-5p in breast cancer. Additionally, the expression of miR-155-5p should be checked in less aggressive stages of breast tumor (Stage-I/II) and compared to stage-III/IV. The same should be done for ovarian cancer samples using publicly available datasets.
3. The authors should provide detailed information about the patient samples used in the study along with the methodology for the methylation analysis and the result.
4. The statement “The fact that curcumin-treated HCC-38 and UACC-3199 cells have lower proliferation rates than untreated cells indicates the oncogenic effect of miR-155-5p in these cells” is not convincing. Functional experiments have not been performed to show that this is the case. For example, the authors in their previous report have shown that curcumin downregulates miR-29b in HCC-38 cells and looked at the possible regulation of TET1 by miR-29b.
5. Apart from IL2RG expression, did the authors check for T-cell functional molecules? Previous study by Wang. et al., JCI, 2022 in mouse have showed increased T-cell influx upon miR-155-5p overexpression.
6. What can be the possible explanation of increased miR-155-5p in both breast cancer and cancer free carriers leading to different downstream effects?
Minor points
1. How does BRCA1 methylation and expression of miR-155-5p compare with the cancer lines tested vs a normal breast cell line like MCF10A?
Minor errors should be fixed.
Author Response
Response to Reviewer 2 Comments Point 1: There are several studies on the involvement of miR-155-5p in breast cancer and other tumors, the authors should emphasize the gap in the field and how this research is an advancement over the existing literature Response 1: We are studying miR-155-5p in peripheral WBCs in cancer-free BRCA1-methylation carriers, which has not been approached previously. Furthermore, all previous studies were about the influence of mutated BRCA1 on miR-155-5p. To the best of our knowledge, our study is the first study about the influence of methylated BRCA1 on miR-155-5p in breast and ovarian cancer. The end of the introduction, lines 88–89, has been changed to "However, the influence of BRCA1 promoter methylation on miR-155-5p expression in BRCA1-methylated breast and ovarian malignancies has not been explored." Point 2: The statement “Overall, these results indicate that high peripheral WBCs miR-155-5p may be tumor suppressive in BC” is not convincing to the reviewer. No functional experiments have been performed. The oncogenic and tumor suppressor role of miR-155-5p in breast cancer have been a topic of debate. Additionally, the number of breast cancer samples used in the study is not large enough for a conclusive statement. The reviewer would advise the authors to use TCGA or other publicly available datasets, segregate patients by BRCA1 status (WT, mutated and methylated) and repeat the analysis. Cox regression and survival analysis should reveal the importance of miR-155-5p in breast cancer. Additionally, the expression of miR-155-5p should be checked in less aggressive stages of breast tumor (Stage-I/II) and compared to stage-III/IV. The same should be done for ovarian cancer samples using publicly available datasets. Response 2: No functional experiments have been performed The conclusion is based on the association of high miR-155-5p levels with favorable prognostic factors. The oncogenic and tumor suppressor role of miR-155-5p in breast cancer have been a topic of debate The statement has been changed to “Overall, these findings suggest that high peripheral WBC miR-155-5p levels may act as a tumor suppressor in BC. Additionally, the number of breast cancer samples used in the study is not large enough for a conclusive statement. The reviewer would advise the authors to use TCGA or other publicly available datasets, segregate patients by BRCA1 status (WT, mutated and methylated) and repeat the analysis. Cox regression and survival analysis should reveal the importance of miR-155-5p in breast cancer. As far as I know, no miR-155-5p BRCA1-methylated related datasets are publicly available for BC. A statement has been added to the discussion section, lines 542-543, “However, further studies are needed to explore the dual functions of these miRs”. Additionally, the expression of miR-155-5p should be checked in less aggressive stages of breast tumor (Stage-I/II) and compared to stage-III/IV. The whole result is based on the comparison between different stages of the disease. The same should be done for ovarian cancer samples using publicly available datasets As far as I know, no miR-155-5p BRCA1-methylated related datasets are publicly available for OC Point 3: The authors should provide detailed information about the patient samples used in the study along with the methodology for the methylation analysis and the result. Response 3: All this information is detailed in reference 40, from which the left-over samples were used Point 4. The statement “The fact that curcumin-treated HCC-38 and UACC-3199 cells have lower proliferation rates than untreated cells indicates the oncogenic effect of miR-155-5p in these cells” is not convincing. Functional experiments have not been performed to show that this is the case. For example, the authors in their previous report have shown that curcumin downregulates miR-29b in HCC-38 cells and looked at the possible regulation of TET1 by miR-29b Response 4. HCC-38 and UACC-3199 are aggressive basal-like cell lines expressing high levels of miR-155-5p and having a high proleferating rates. The fact that curcumin treatment reduces the level of miR-155-5p together with the proliferation rate indicates the oncogenic effect of miR-155 in these cells. Point 5. Apart from IL2RG expression, did the authors check for T-cell functional molecules? Previous study by Wang. et al., JCI, 2022 in mouse have showed increased T-cell influx upon miR-155-5p overexpression Response 5. No, we did not. Point 6. What can be the possible explanation of increased miR-155-5p in both breast cancer and cancer free carriers leading to different downstream effects? Response 6. In our previous study, Ref 40, we showed that miR-126 was highly expressed in WBCs in both breast cancer and cancer-free carriers. However, plasma miR126, which has been reported to differentiate patients with cancer from controls, was downregulated in BC but not in the carriers, indicating the abnormality of the carriers, who are still cancer-free. Similar to miR-126, serum miR-155 in breast cancer patients is reported to serve as a non-invasive diagnostic biomarker Ref [34-36]. This could be a possible explanation of the increased miR-155-5p in both breast cancer and cancer-free carriers with different downstream effects. Minor points 1. How does BRCA1 methylation and expression of miR-155-5p compare with the cancer lines tested vs a normal breast cell line like MCF10A? Response. We did not look into the expression of miR-155-5p in the normal breast cell lines like MCF10A, but I assume it would be related to the level of BRCA1 protein in these cells. English editing is done
